## RESEARCH ARTICLE

# Caspar modulates primordial germ cell fate both in an Oskar-dependent and Oskar-independent manner

Subhradip Das, Adheena Elsa Roy, Kanika, Girish Deshpande* and Girish S. Ratnaparkhi*

## ABSTRACT

Primordial germ cell (PGC) formation and specification is a fundamental conserved process as PGCs are the progenitors of germline stem cells (GSCs). In *Drosophila melanogaster*, maternally deposited Oskar (Osk) and centrosome dynamics are two independent determinants of PGC fate. Caspar, *Drosophila* homolog of Fas-associated factor 1 (FAF1), promotes PGC formation/specification and maintains the PGC count by modulating both the Osk levels and centrosome function. Consistently, *casp^{lof}* PGCs display reduction and inefficient release/transmission of germ plasm. Defective centrosome migration and behavior are evident even prior to PGC formation engineered by Osk and its targets. Taken together with the inability of Osk to regulate nuclear and centrosome migration, our data demonstrate that Casp encodes a novel bi-modal regulator of PGC fate as it controls Osk levels likely by downregulating translational repressor, Smaug (Smg) and also influences nuclear/centrosome migration during early mitotic nuclear division cycles (NCs 6-9), which are Osk-independent. We discuss dual functionality of Casp vis-à-vis germline/soma segregation as it helps acquire both the PGCs and the surrounding soma their individual identities.

KEY WORDS: Pole cells, PGC, Oskar, Centrosome, Migration, Germplasm

## INTRODUCTION

Primordial germ cells (PGCs), the progenitors of the germline stem cells (GSCs), acquire the stem cell identity over the course of development (Chen et al., 2025; Extavour and Akam, 2003; Nakamura and Seydoux, 2008). PGCs are endowed with distinguishing traits as compared to the surrounding soma and maternal factors underlie the establishment and maintenance of the PGC identity. Specifically, PGCs are translationally competent but transcriptionally quiescent, display specialized chromatin architecture and have limited mitotic potential. Together, these functional attributes restrict the total number of PGCs and enable them to acquire the GSC fate (Santos and Lehmann, 2004; Strome and Lehmann, 2007). In sum, germline/soma distinction is initiated when PGCs form, and is maintained throughout the life cycle (Ewen-Campen et al., 2010).

Department of Biology, Indian Institute of Science Education & Research, Pune 411008, India.

*Authors for correspondence (girish@iiserpune.ac.in; gdeshpande@iiserpune.ac.in)

S.D., 0009-0002-4553-966X; A.E.R., 0009-0008-6000-9572; K., 0000-0002-1034-8655; G.D., 0000-0002-5200-7090; G.S.R., 0000-0001-7615-3140

Extensive analysis of PGC formation and specification in *Drosophila melanogaster* has revealed critical functions of maternally inherited specialized cytoplasm or germ plasm in these processes. Germ plasm enriched in the PGC determinants is cortically anchored at the posterior pole in the fertilized eggs (Chen et al., 2025; Wang and Seydoux, 2013). Early experiments showed that transplantation of pole plasm at an ectopic location is sufficient to form pole cells at the site of injection. This pioneering observation demonstrated that germ plasm contains essential PGC determinants (Illmensee and Mahowald, 1974, 1976; Mahowald, 2001).

Follow-up studies identified Oskar (Osk) as a critical determinant of the PGC fate. Evidently, maternal loss or gain of *osk* leads to corresponding reduction or increase in the total PGC count. Critically, ectopic expression of *osk* is sufficient to transform somatic cells into functional PGCs as Osk can engineer germ plasm assembly (Ephrussi and Lehmann, 1992; Kim-Ha et al., 1991; Lehmann, 2016).

While Osk is capable of recruiting the PGC determinants, individual downstream components display a division of labor and modulate defined aspects of PGC fate and/or PGC count. For instance, silencing of Pol-II dependent transcription is mediated by Polar granule component (Pgc), Nanos (Nos) and Germ cell less (Gcl), which have overlapping but unique functions (Asaoka-Taguchi et al., 1999; Deshpande et al., 2004, 1999; Hanyu-Nakamura et al., 2008; Leatherman et al., 2002; Martinho et al., 2004). However, among the three, formation of PGCs is regulated only by *gcl*.

Proper formation of functional PGCs is also dependent on the centrosomal migration. A fertilized embryo undergoes 13 serial mitotic nuclear division cycles (NCs) before cellularization occurs during the 14th NC. The first nine NCs happen in the center of the embryo before nuclei migrate towards the periphery. Just prior to the cortical migration, a few nuclei precociously invade the germ plasm along with the associated centrosomes (reviewed by Blake-Hedges and Megraw, 2019; Fu et al., 2015). Elegant studies by Raff and Glover demonstrated that migration of the centrosomes into the germ plasm, rather than the nuclei, is crucial for the formation of PGCs (Raff and Glover, 1989).

Subsequent live imaging analysis established that entry of centrosomes induces the release of the cortically anchored germ plasm (Lerit and Gavis, 2011; Lerit et al., 2017). Moreover, astral microtubules emanating from the posterior centrosomes are involved in the even distribution of germ plasm among the dividing early PGC nuclei. Consistently, centrosome components (e.g. centrosomin) are essential for proper PGC formation. Moreover, severe reduction in PGC numbers observed in *gcl* mutant embryos was correlated with aberrant centrosome separation and inequitable distribution of germ plasm components between dividing PGCs (Lerit and Gavis, 2011; Lerit et al., 2017).

We recently reported a novel maternal requirement of Caspar (Casp) protein in proper formation and specification of PGCs (Das et al., 2024). Casp encodes *Drosophila* homolog of the mammalian

Biology Open

Fas-associated factor 1 (FAF1). It is a member of the tumor necrosis factor receptor family and is pro-apoptotic. FAF1 is implicated as a negative regulator of immune signaling as it influences the import of RelA/NF-κB (Min-Young et al., 2004; Park et al., 2007). Similarly, Kim et al. (2006) uncovered a role for *casp* in flies where it regulates Relish/NF-κB import, presumably by regulating Dredd, which has the ability to cleave Relish during immune signaling.

Our recent studies focused on embryonic functions of maternally deposited *casp*. A significant proportion of *casp^{lof}* embryos at blastoderm stage display defective cytoskeletal network and centrosome aberrations leading to gastrulation failure (Das et al., 2024). These traits resemble embryonic phenotypes of mutants in several other protein components involved in mid-blastula transition (MBT) (Harrison and Eisen, 2015; Harrison et al., 2023; Vastenhouw et al., 2019). Interestingly, Casp is also expressed strongly in the PGCs. Moreover, consistent with the ability of Casp to regulate Osk levels, 'loss' or 'gain' of *casp* results in reduction or increase in total PGC count, respectively. Accordingly, pole cell formation is modulated by Casp and the total number of pole buds is directly proportional to the level of Casp (Das et al., 2024). Subsequent analysis of several known regulators of MBT enabled us to identify the translational repressor Smaug (Smg) as a potential target of Casp. Smg was recently shown to influence Osk levels negatively, and thus, embryos derived from *smg* mutant mothers display elevated numbers of PGCs (Siddiqui et al., 2024). Taken together, our results suggested that Casp may mitigate this effect by keeping Smg levels in check.

While novel, these findings raised several intriguing questions: What is the eventual fate of *casp^{lof}* PGCs? Do they display any characteristic phenotypes during later stages of embryogenesis, and if so, is the aberrant behavior reminiscent of any mutations in the known maternal genes involved in PGC development or migration? Most importantly, does Casp regulate germline/soma segregation simply due to its influence on Osk levels? Or does it also act via its effect on nuclear and centrosomal migration during early NCs? If it does, how the centrosomes associated with such nuclei influence the transport of posteriorly anchored germ plasm? Is the ability of Casp to modulate centrosome behavior an important contributing factor underlying defective PGC formation in *casp^{lof}* embryos?

Here we address several of these questions. Our findings suggest that maternally deposited *casp* is a novel dual regulator capable of influencing Osk levels and nuclear migration as well as centrosome dynamics in pre-syncytial blastoderm embryos. Consequently, it can control equitable germ plasm distribution between dividing PGCs to determine their total count and fate. We discuss these observations in light of two distinct criteria with regards to PGC specification: 1) Osk-dependent germ plasm assembly and, 2) Osk-independent centrosome migration, which allows their entry into germ plasm, leading to its carefully calibrated release. Even distribution of adequate levels of germ plasm among embryonic PGCs depends on both these determinants underlying proper formation and specification of PGCs.

## RESULTS
### Analysis of PGC count in the *casp^{lof}* embryos during embryogenesis
We have previously shown (Das et al., 2024) that Casp is required for embryonic pole bud formation and its subsequent mitotic divisions (Das et al., 2024). Here we investigate the fate of the PGCs from the *casp^{lof}* embryos during germ-band elongation and retraction (Fig. 1).

To examine if the PGC count decreases as gastrulation progresses, we stained *casp^{lof}* embryos and control samples using a PGC specific marker, anti-Vasa antibodies. To assess the influence of *casp* on the

PGCs during developmental progression, we collected embryos spanning stages 8-11 and stages 12-15 (Fig. 1A) and compared the total number of Vasa positive cells, i.e. PGCs (Fig. 1B; Fig. S1A,B).

First, we analyzed embryos between stages 8-11 of different genotypes, including *casp^{lof}*, *casp^{OE}* (maternal overexpression of *casp*) and control (Fig. 1B). Expectedly, the total number of PGCs in the mutant embryos is significantly lower (~8) than the control (~30 PGCs). Importantly, the mutant phenotype is almost exclusively maternal, as the PGC numbers are diminished significantly only when the mothers are homozygous mutant. By contrast, embryos derived from the control females mated with the homozygous mutant males displayed similar number of PGCs as control embryos (25-30 PGCs).

Next, we assessed if the total number of PGCs reduced between stages 8-11 to stages 12-15 (Fig. 1C). As expected, in controls, PGCs gradually declined to 20 per embryo as several PGCs are lost while they undergo directed migration prior to gonad coalescence (Fig. 1C; Fig. S1A,B). Interestingly, for *casp^{lof}* the number of PGCs stayed constant (6-8 PGCs; Fig. 1C; Fig. S1A,B). Moreover, this phenotypic trait is also strictly correlated with the maternal genotype. Altogether, these data showed that *casp^{lof}* embryos, display substantially lower PGC counts till stage 5, whereas in later stages of embryogenesis the decrease is less pronounced, unlike in controls (Fig. S1A,B), suggesting that the requirement of Casp activity for the continued survival of PGCs is perhaps less acute.

For *casp^{OE}* (nosGal4>UAS casp) embryos, we observed a slight elevation in the total number of PGCs in embryos (stages 8-11). Notably, this increase was not as pronounced as that observed in younger, i.e. stage 5 embryos (~40 PGCs; Das et al., 2024). Importantly, resulting embryos (*casp^{OE}*) exhibited reduction in PGC numbers between stages 8-11, similar to the decrease observed in control embryos.

Interestingly embryos of *nosGal4/UAS casp* females at stage 12-15 (Fig. 1C; Fig. S1A,B) display higher numbers of PGCs, supporting the conclusion that presence of excess Casp can protect PGCs from apoptotic or phagocytic loss to a certain extent. Taken together this analysis indicated that: 1) germ cell loss observed in *casp^{lof}* embryos is strictly maternal. 2) Reduction in total germ cell count observed in *casp^{lof}* embryos at mid-embryogenesis does not exacerbate appreciably in older (stage 12-15) embryos, and unlike the sharp decline in wild-type embryos. 3) PGC-specific expression of *casp* renders pole cells partially resistant to progressive loss.

### Phenotypic similarities between *casp^{lof}* and *nos^{m−}* PGCs
While the PGC counts are reduced in *casp^{lof}* embryos at stage 5, almost all of these cells survive until stage 11-12. We examined how such 'survivor' mutant PGCs behave during mid and late embryogenesis. We compared their migratory behavior with control PGCs that typically undergo directed migration to form a primitive embryonic gonad (reviewed in Barton et al., 2016; Grimaldi and Raz, 2020; McDonald and Montell, 2005; Tarbashevich and Raz, 2010).

As gastrulation proceeds between stages 6-8, PGCs specified at the posterior, initially migrate over the external dorsal surface, are internalized upon midgut invagination and move along the gut cavity as germ band extension proceeds. At stage 9, PGCs individualize at the blind end of the hindgut to undergo trans-epithelial migration across the gut membrane. By stage 10, PGCs move along the dorsal surface of the gut before splitting into two similar sized groups. While not physically attached, PGCs migrate in a coordinated manner across the mesoderm (stages 11-13) towards the somatic gonadal precursor cells (SGPs) on either side of the embryo, ultimately giving rise to primitive embryonic gonads.

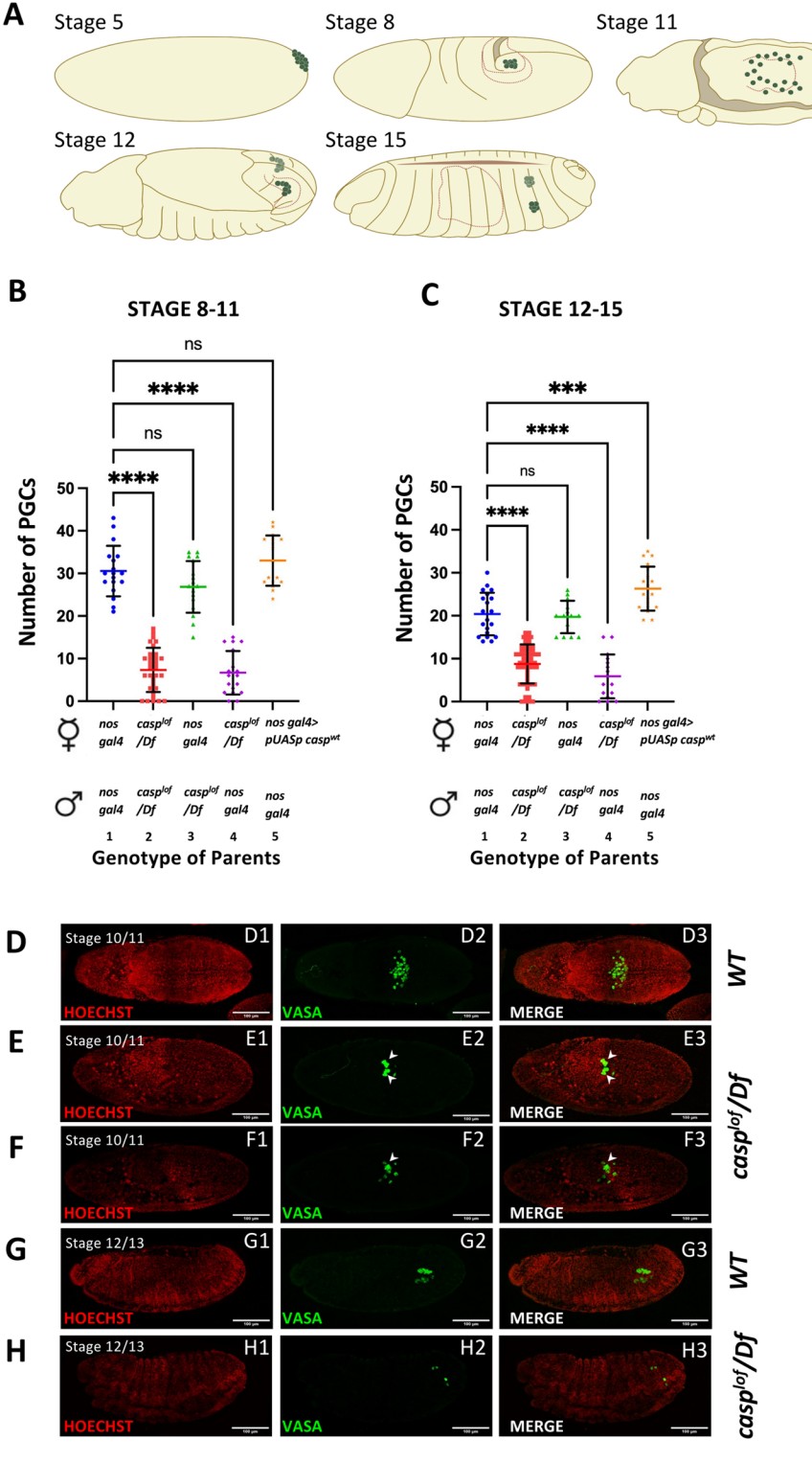

**Fig. 1. Defects in PGC migration in *casp^lof* embryos.** (A) Schematic for different stages of PGC specification and migration in *Drosophila*. The developmental events are described in the main text. (B) At stages 8-11, embryos laid by *casp^lof*/Df mothers display ~8 PGCs irrespective of paternal genotype (as opposed to ~27 PGCs when fathers are *casp^lof*/Df and ~30 PGCs when both parents are control *nos gal4*). (C) The typical reduction of PGCs seen in control samples through stages 12-15 was not apparent in the progeny of *casp^lof*/Df mothers, which was stable at 7-9 PGCs (panels B versus C). Control females mated with *casp^lof*/Df fathers produced progeny with ~20 PGCs, similar to control. Maternal overexpression of *casp* showed a slight elevation in stages 8-11 but a prominent PGC count (~27) in stages 12-15 (C). In panels B and C, the parental genotype is listed on the X-axis with total number of PGCs plotted along the Y-axis. Each point on the graph represents one embryo, with 14-50 embryos represented (see Fig. S1A) per genotype, per stage, for panels B and C. Ordinary one-way ANOVA/unpaired *t*-test, ****$P<0.0001$; ns, not significant. (D-H) Confocal images of the whole embryos of stage 10/11 (D-F) and stage 12/13 (G,H) laid by *nos gal4* (D,G) and *casp^lof*/Df (E,F,H) mothers. The embryos were immunostained with Vasa (green) antibody. Hoechst marks the nuclei (red). PGCs from a significant subset of embryos laid by *casp^lof*/Df mothers remain as clusters (albeit post-individualization) (see arrowheads in panels E2-3 and F2-3) unlike those in controls.

*casp^lof* PGCs, while fewer in number, displayed near normal pattern of migration until reaching the end of the hindgut. Similar to control, the mutant PGCs individualize and traverse the midgut pocket by stage 9. Unlike in the controls (Fig. 1D), the mutant PGCs appeared to linger on the dorsal surface of the gut ultimately forming small clusters (Fig. 1E), which remain in the middle of the respective embryos, giving an appearance of ectopic 'pseudo-gonads'. Other relatively minor defects included scattering of PGCs or mislocalization of small clusters across the posterior half.

While several maternal gene products control PGC migration in the *Drosophila* embryo, the specific phenotypes seen in *casp^lof* embryos are partially reminiscent of *nos^m−* (Deshpande et al., 1999; Forbes and Lehmann, 1998; Kobayashi et al., 1996). Interestingly unlike *casp*, Nos protein does not contribute to the formation of early pole buds/cells. However, after exiting the midgut, *nos^m−* PGCs remain stationary, forming variably sized clumps without migrating across the mesoderm. The requirement of Nos function is germ cell autonomous as embryos obtained from females carrying *nos^−hb^−* germline clones

exhibited identical PGC migration defects. By contrast, patterning defects induced by the loss of maternal *nos* function are significantly rescued by simultaneous removal of maternal *hunchback* (*hb*). That *nos* function is germ-cell autonomous was conclusively established by the pole cell transplantation experiments (Kobayashi et al., 1996). Thus, we sought to explore if maternal loss of *casp* can influence *nos* function adversely.

### *nos* transcripts show reduction and aberrant localization in *casp^{lof}* embryos

While striking, the phenotypic similarity between PGC migration defects in *nos*^{m−}*hb*^{m−} and *casp^{lof}* embryos is not altogether surprising. We have shown that Casp protein is necessary for downregulating Smg in the early embryos, which, in turn, functions as a negative regulator

of both *osk* and *nos* RNA translation and localization. Previous studies have also established that Osk is a positive regulator of *nos* localization and translation (Ephrussi et al., 1991; Smith et al., 1992). Prompted by these observations, we examined *nos* transcript levels in *casp^{lof}* and *nos^{OE}* embryos by performing RNA *in situ* hybridizations. As can be seen in Fig. 2A1-3, in control embryos *nos*-specific signal is seen as a posterior cap while trace amounts of *nos* RNA are spread through the rest of the embryo.

The intensity and spread of *nos*-specific signal at the posterior cortex diminishes in *casp^{lof}* embryos (Fig. 2B1-3), while it is elevated in *nos^{OE}* embryos (Fig. 2D,E). The RNA levels at the posterior are reduced by half in the embryos derived from mothers compromised of *casp*, whereas they nearly double in *nos^{OE}* embryos.

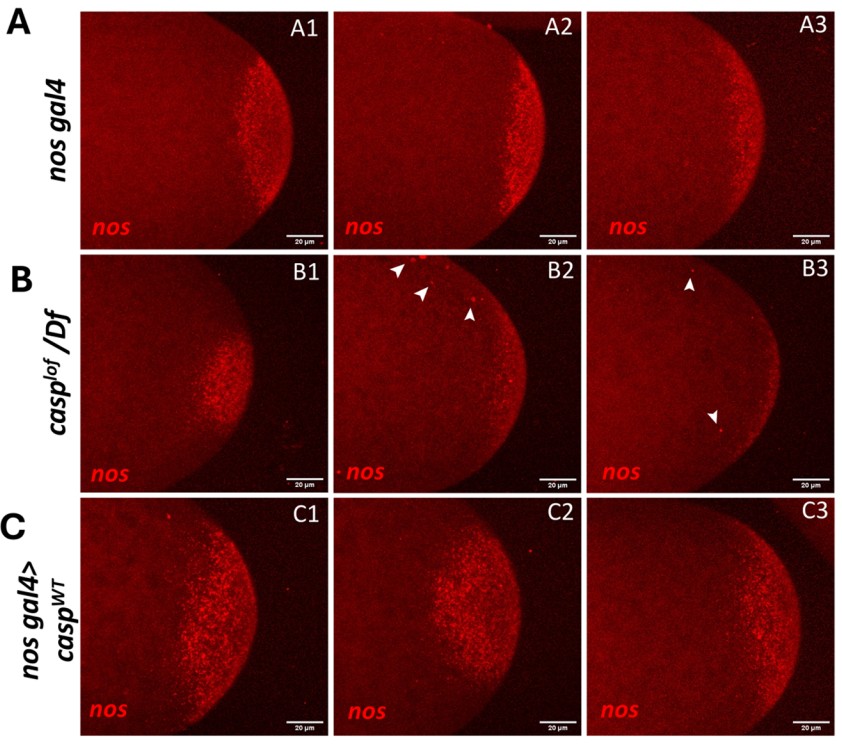

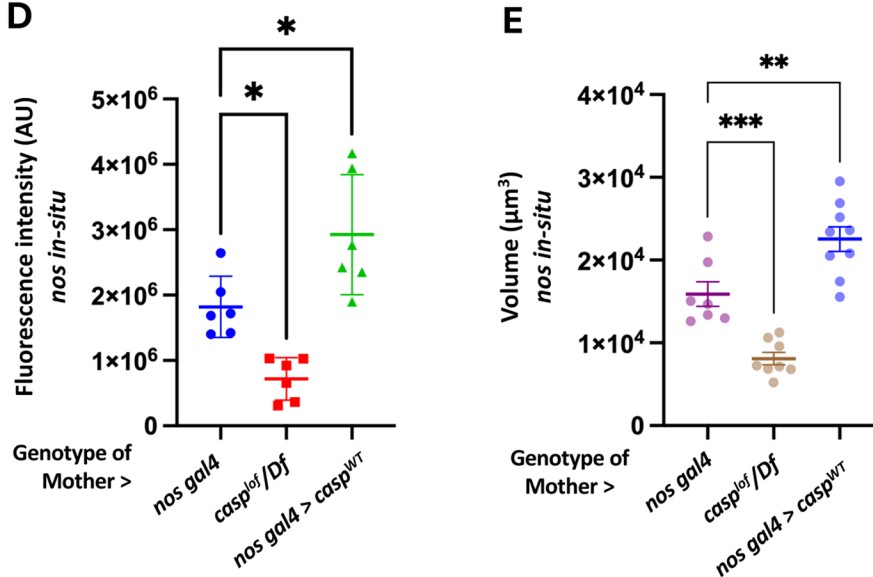

**Fig. 2. Casp regulates *nos* mRNA levels and localization.** (A-C) RNA *in situ* hybridizations were performed using *nos* specific probe on stage (1/2) embryos derived from *nos gal4*, *casp^{lof}/Df*, and *casp^{OE}* mothers, and detected and quantitated as described in the Materials and Methods. Panels 1-3 are replicates for each genotype, highlighting the variable changes in *nos* mRNA levels. In comparison to the *nos gal4*, which served as control sample (A1-A3), *nos* mRNA levels seem discernibly reduced in *casp^{lof}/Df* (B1-B3). By contrast, the levels are elevated in *casp^{OE}* embryos (C1-C3). In addition to the reduced levels of *nos* mRNA in *casp^{lof}/Df* embryos, *nos* transcripts appear as ectopic puncta away from the posterior cortex (arrowheads in B2, B3). (D) Fluorescence intensity quantification of *nos* mRNA per embryo plotted for *nos gal4*, *casp^{lof}/Df*, and *casp^{OE}* embryos. (E) Volume (µm³) measurements of *nos* mRNA per embryo at the posterior cortex of all three genotypes (*nos gal4*, *casp^{lof}/Df*, and *nos gal4>casp^{WT}*). One-way ANOVA, *$P<0.05$, **$P<0.01$, ***$P<0.001$.

In addition to decrease in the levels, we observed that *casp^{lof}* embryos display somewhat aberrant localization of *nos* RNA. On several occasions, *nos* RNA accumulated as ectopic puncta in the posterior third of the *casp^{lof}* embryos. By contrast, in the control embryos, such accumulation away from the posterior cortex was rarely observed. Smg is a negative regulator of *nos* localization and translation (Dahanukar et al., 1999; Smibert et al., 1999). As we previously reported, Smg levels are elevated in *casp^{lof}* embryos, and likely contribute to the diminished levels of *nos* in mutant embryos. Intriguingly *casp^{OE}* embryos (Fig. 2C), with excess *nos* RNA at the posterior cortex, display modest mislocalization as compared to the control embryos, suggesting that the elevated levels of RNA exist as more densely packed RNA granules.

### Vasa protein spreads in the posterior half of the *casp^{lof}* embryos

Mitigating Osk levels/activity influences germ plasm assembly adversely, ultimately resulting in a reduction of the total number of PGCs. *casp^{lof}* embryos show diminished Osk and PGC counts. We did not, however, see any significant change in localization of Osk itself in *casp^{lof}* embryos. By contrast, *nos*, one of the important targets of Osk, is not simply reduced in *casp^{lof}* embryos but *nos* transcripts are mislocalized in the form of ectopic puncta away from the posterior cortex. We thus wondered if this effect is unique to *nos* or other pole plasm components behave similarly. To examine this possibility, we co-immunostained *casp^{lof}* embryos with anti-Osk and anti-Vasa antibodies (Fig. 3). Vasa, another important component of germ granules, is a canonical target of Osk.

Expectedly both Osk and Vas proteins are completely coincident at the posterior cortex in the control embryos (Fig. 3A1-3). *Casp^{OE}* embryos (Fig. 3C1-3) also display similar coincident cortical localization of Osk and Vas with trace amounts of Vas in the posterior. Interestingly, however, in *casp^{lof}* embryos, in addition to a complete overlap at the posterior cortex, readily detectable levels of Vas protein are spread in the posterior half of the mutant embryos (Fig. 3B1-3).

To quantify the spread of Vas, we plotted the signal intensity across the length of the embryo using both control and *casp^{lof}* samples (Fig. 3E,F). Again, in the control embryos, Vasa is mostly detected in the posterior quarter. In contrast, in the *casp^{lof}* embryos, peak levels of Vas are observed away from the posterior cortex indicating aberrant farther spread. Consequently, considerable levels of Vasa are observed in the posterior half indicating inefficient anchoring akin to *nos* RNA (Fig. 3E,F).

### Inappropriate release and spread of germ plasm components correlate with aberrant centrosomes

Our data suggested that two critical germ plasm components, *nos* RNA and Vas protein, display inappropriate spread and variable ectopic accumulation in the *casp^{lof}* embryos. Together, these data suggested that either the germ plasm is not anchored to the posterior cortex properly or it undergoes inappropriate release. Our data (Das et al., 2024; this work) showed that while Osk levels change both in *casp^{lof}* and *casp^{OE}* embryos (Fig. 3), Osk protein remains tightly associated with the posterior cortex. As anchoring of Osk seemed relatively normal, unlike *nos* RNA and Vas protein, we decided to explore if germ plasm components are inappropriately released from the posterior cortex resulting in unusual spread.

The controlled release of the germ plasm is triggered by the entry of the centrosomes around NC 9, leading to pole bud formation during NC 10. Microtubules emanating from the centrosomes allow

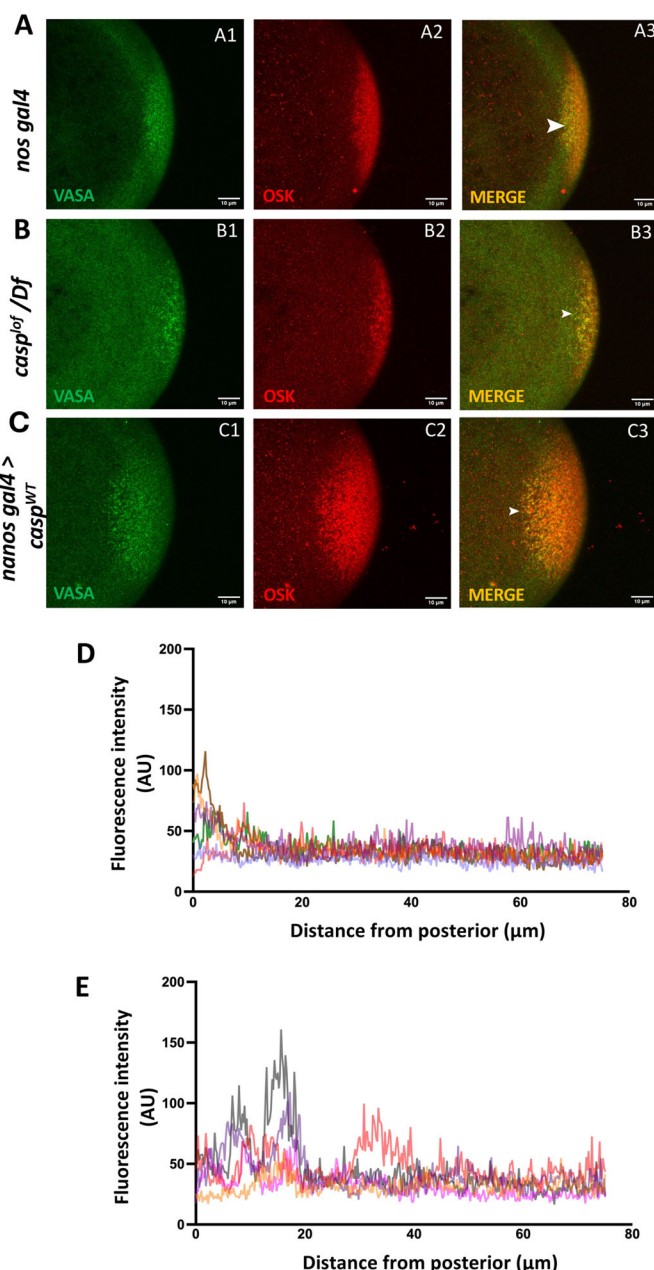

**Fig. 3. Casp modulates levels and localization of the pole plasm components.** (A-C) Confocal sections of pre-syncytial blastoderm (0-1.5 h-old) embryos derived from *nos gal4*, *casp^{lof}/Df*, and *casp^{OE}* females immunostained with Osk and Vasa antibodies. A3, B3, C3 are the merged panels showing the colocalization of Osk and Vasa at the posterior cortex (see arrowheads). (D,E) Fluorescence intensity of Vasa measured starting from the posterior cortex till 75 µm to the anterior for the control (D), and *casp^{lof}/Df* (E) embryos. *casp^{lof}/Df* embryos show spread of Vasa towards the anterior as compared to the control.

for the transport of germ plasm components. While the pole buds divide, the germ plasm is evenly partitioned among the dividing pole buds and/or cells using the astral microtubules. We thus wondered whether the *casp^{lof}* buds display even partitioning of the germ plasm components and examined the levels of Vasa between dividing pole buds or PGCs. We also marked the centrosomes using γ-tubulin antibodies. In the control embryos at NC 10 (Fig. 4A1-4), Vasa is uniformly distributed among the daughter cells and centrosomes are at the opposite poles. Moreover, somatic nuclei

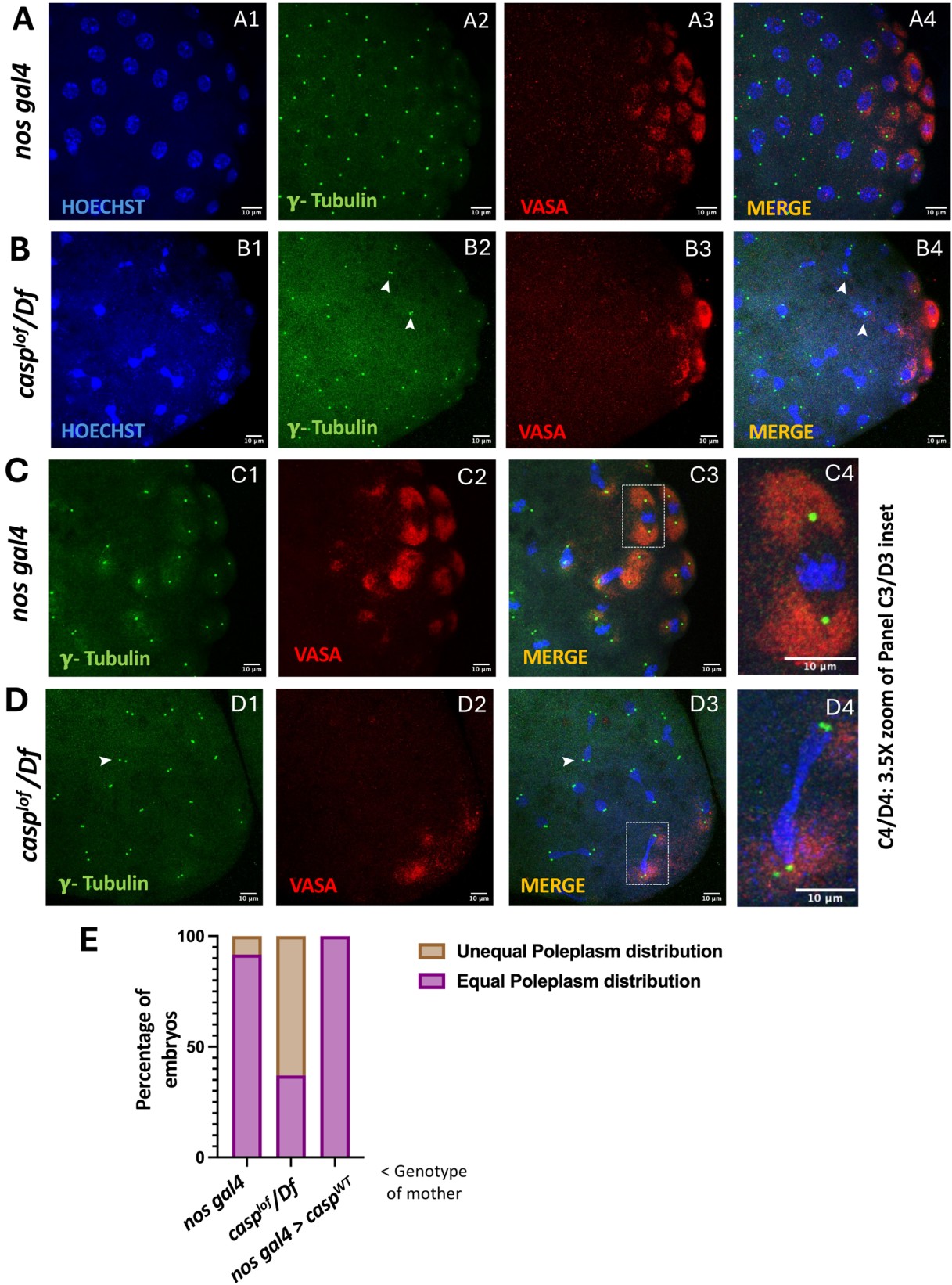

**Fig. 4. Even distribution of pole plasm components between the dividing pole buds is disrupted in *casp^lof*/d.f. embryos.** (A-D) Nuclear cycle 10 and 9 embryos laid by *nos gal4* (A,C) and *casp^lof*/Df (B,D) mothers immunostained with γ-tubulin (centrosome marker) and Vasa (PGC marker). Hoechst labels the nuclear DNA. Defective centrosomes in the *casp^lof*/Df embryos are highlighted using arrowheads (B2,B4). The pole plasm distribution in both the genotypes is clearly seen by comparison of panels C4 and D4, which are magnified (3.5 X) versions of the insets from the panels C3, and D3, respectively. Consistent with the centrosome defects, *casp^lof*/Df embryos (D4) show non-uniform distribution of pole plasm components, as compared to the control (C4). (E) Bar chart representation of the percentage of embryos showing either equal or unequal distribution of pole plasm for *nos gal4*, *casp^lof*/Df, and *casp^OE* embryos.

as well as the associated centrosomes are regularly spaced. In contrast, pole buds from $casp^{lof}$ embryos (Fig. 4B1-4) exhibit reduced levels of Vasa, which is unevenly partitioned (Fig. 4B3). Furthermore, the nuclear division is asynchronous, with uneven distribution of centrosomes in the somatic compartment. The uneven distribution of Vas among the daughter nuclei could result from centrosome aberrations such as incomplete separation and/or delayed migration among the dividing pole buds.

To trace it back to earlier NCs we analyzed centrosomes from cycle 8-9 embryos when pole cell nuclei along with the associated centrosomes enter the germ plasm to induce its release (Fig. 4C1-3). Again, the somatic nuclei and associated centrosomes from control embryos exhibit regular arrangement and the germ plasm is evenly partitioned among the daughter pole buds (Fig. 4C4).

In sharp contrast, the $casp^{lof}$ pole buds display inequitable distribution of Vasa, which is accompanied by aberrant centrosome divisions. Consequently, one pole bud seems to have acquired substantially higher levels of Vasa compared to the other daughter pole bud Fig. 5, (compare panels C3 and D3 and the respective insets; Fig. 5C4, D4). This feature is quantified in (Fig. 4E). γ-tubulin specific staining in the pole buds is weaker and even in the somatic nuclei the centrosomes are non-uniformly distributed as their separation seems incomplete or slower.

### Centrosome defects correlate with reduced levels of Gcl in $casp^{lof}$ embryos

While the even distribution of germ plasm between dividing PGCs depends on centrosomes, how the centrosome behavior is precisely modulated in this context is not completely understood. Gcl is another germ plasm component that regulates PGC count (Jongens et al., 1994, 1992) as it is required for pole bud and PGC formation (Cinalli and Lehmann, 2013; Jongens et al., 1994). Efficient centrosome separation post-duplication relies on Gcl activity (Lerit et al., 2017). Loss of $gcl$ results in improperly elaborated astral microtubules leading to non-uniform distribution of germ plasm, which is correlated with PGC loss. Given the similarity in the phenotypes between the PGCs from $gcl^{m-}$ and $casp^{lof}$ embryos, we analyzed Gcl protein levels and distribution in $casp^{lof}$ embryos. $casp^{lof}$ and control embryos (0-3 h old) were stained with anti-Gcl and anti-Vas antibodies and counterstained with a DNA dye, Hoechst (Fig. 5). In control embryos, both Gcl and Vas proteins are enriched in PGCs (Fig. 5A1-4). Our data confirmed that Gcl is associated with the nuclear envelope and is also part of the nucleoplasm, while Vas is cytoplasmic. As expected, levels of both Gcl and Vas are reduced in $casp^{lof}$ PGCs (Fig. 5B1-4). In the $casp^{lof}$ pole buds and PGCs, Gcl protein is not simply reduced but its localization is also altered. In the mutant pole buds and PGCs, Gcl protein is readily detectable in the nucleoplasm as compared to the nuclear envelope (see Fig. 5B,C; quantitation of Gcl reduction shown in Fig. 5E). Altogether, these data corroborate the earlier conclusion that both nuclear envelope association and nucleoplasmic localization of Gcl have distinct functions and proper PGC formation may depend on Gcl being localized to the nuclear envelope (Robertson et al., 1999).

As in the case of $gcl^{m-}$, in the $casp^{lof}$ embryos pole bud formation is affected. In several instances, the mutant pole buds are not fully extruded or pinched off from the surface (compare Fig. 5, panel A with panel B, showing unsuccessful extrusion of pole buds marked by the presence of Gcl). Moreover, nuclear DNA in the mutant pole buds is severely fragmented compared to control (see magnified images of a single PGC from control sample with the $casp^{lof}$ PGC shown in Fig. 5C, see below).

### Budding of pole cells in $casp^{lof}$ embryos is affected due to improper contractile ring formation and incomplete closure

The discernible defects in pole bud formation in $casp^{lof}$ embryos prompted us to analyze the bud extrusion process further. Precocious cellularization of pole buds involves a basal actin ring constriction that depends on components of the actin cytoskeleton including *Drosophila* Septin, Peanut (Pnut), Anillin, and Rhogef2, which accumulate at the base of the pole buds (reviewed in Chen et al., 2025). The same proteins are found at the newly formed cell membranes in the soma. The specific localization and resulting accumulation of these proteins is functionally significant as PGC formation is affected by individual mutations coding either these genes or in the regulators that engineer their proper localization.

Staining of the control embryos at stage 3-4 with anti-Pnt (red) and anti-Vasa (green) antibodies revealed the basal accumulation of Pnut that separates pole buds from the rest of the embryo (Fig. 6A,B). The process of pole cell individualization and formation is abnormal in $casp^{lof}$ embryos (Fig. 6C,D). At the time of protrusion of the pole buds, the aberrant pattern is readily detectable. In the control embryos, the completion of pole bud formation is marked by a thin band of Pnut-specific staining at the base (Fig. 6A2-4,B2-4). At this stage the fully extruded pole buds lie on the external surface, which marks the end of the process. Concomitantly, the cell membranes of newly formed pole cell are weakly labeled by Pnut antibodies.

In contrast the process of pole bud formation in $casp^{lof}$ embryos is incomplete and on several instances the buds are not fully extruded and instead they remain either partially or completely embedded in the embryonic surface (Fig. 6C2-4,D2-4). Pole buds from $casp^{lof}$ embryos differ from wild type in the levels of Pnut staining. In most cases the band of Pnut protein separating the pole cells from the soma is nearly absent and very weakly stained compared to the control embryos (compare Fig. 6C,D to Fig. 6A,B). These observations show that unlike in control embryos, pole buds from $casp^{lof}$ embryos are unable to pinch off on several occasions.

### Nuclear and centrosomal defects in $casp^{lof}$ embryos are observed prior to pole bud formation

Thus far, the aberrant phenotypes observed in syncytial blastoderm $casp^{lof}$ embryos include mislocalization and/or reduction in germ plasm components, defective budding, and diminished PGC count. Many of these phenotypes are reminiscent of $gcl^{m-}$ embryos. Furthermore, PGC migration defects seen in older $casp^{lof}$ embryos resemble that of $nos^{m-}$. Consistently, Gcl and $nos$ levels are also reduced in the $casp^{lof}$ PGCs. Altogether, all of the PGC-specific defects can be readily attributed to reduction in one or more components of germ plasm that are assembled downstream of *Osk* activity. We thus wondered if ability of Casp to downregulate Osk levels/activity is the sole factor that contributes to PGC-specific aberrations due to maternal loss of $casp$. Thus, we decided to revisit our observations concerning nuclear and centrosomal migration defects seen in syncytial blastoderm embryos. To assess if we could trace the aberrant behavior to earlier stages, we stained presyncytial blastoderm $casp^{lof}$ embryos along with similarly staged control embryos with a DNA dye, Hoechst. As can be seen from Fig. 7, the nuclei from $casp^{lof}$ presyncytial embryos (between NC 6-10) show several different defects, including fragmentation of nuclei leading to fragmentation of DNA, irregular spacing, mitotic asynchrony, etc. Interestingly, in addition to the centrosome aberrations, in several instances nuclei also display uneven chromosome segregation possibly due to kinetochore defects. In an independent experiment similar defects were also displayed by the centrosomes associated with the nuclei. As

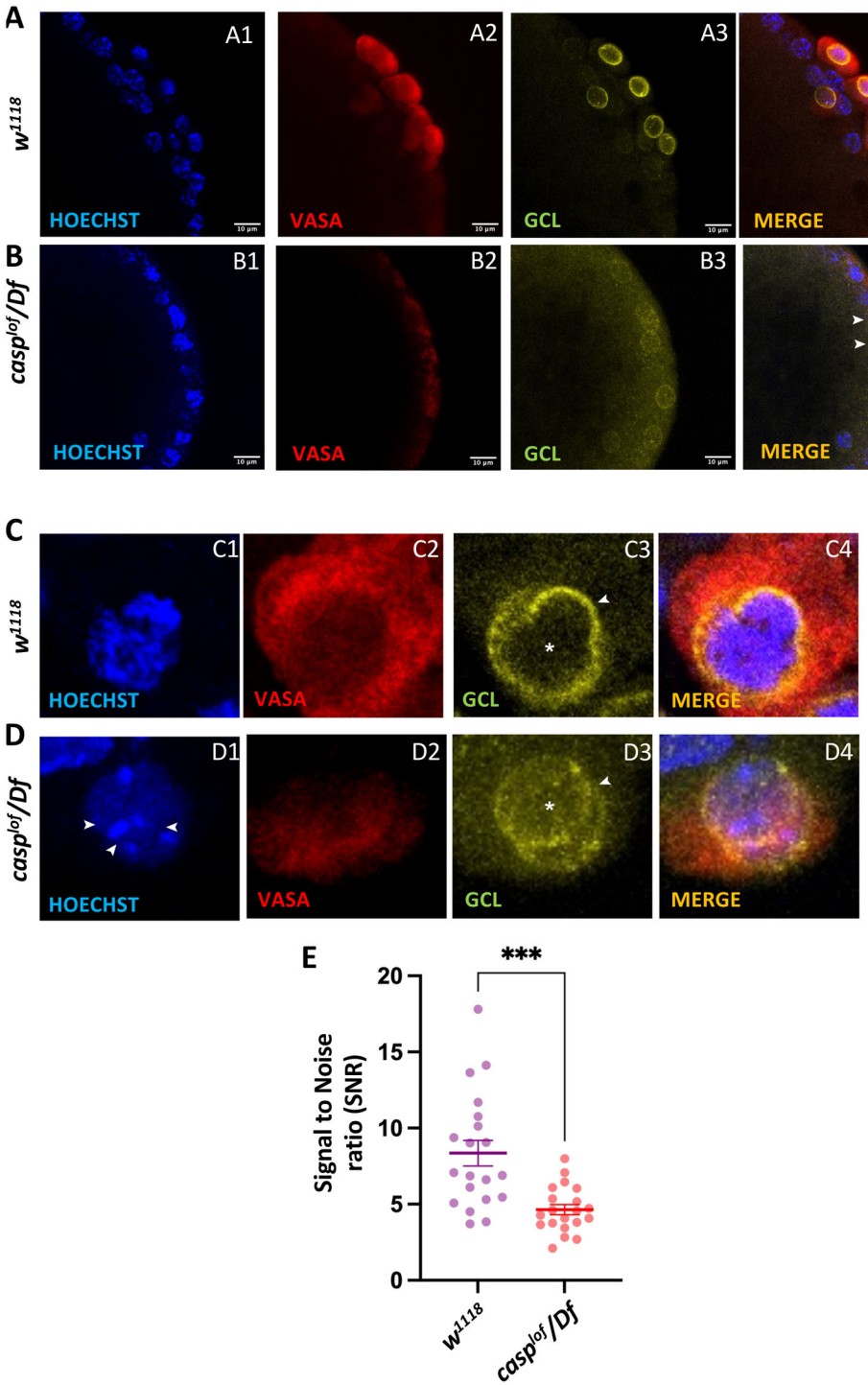

**Fig. 5. *casp^lof^/Df* pole buds display reduced levels and aberrant subcellular localization of Gcl.**
(A-D) Nuclear cycle (10-11) embryos laid by *w^1118^* (A,C) and *casp^lof^/Df* (B,D) mothers immunostained with Vasa, and Gcl antibodies and counterstained with Hoechst (DNA dye). Confocal images of both the control and mutant pole buds are shown in panels A-D. Two defects are readily apparent by comparison of mutant pole buds with the control. In control pole buds, Gcl is readily seen associated with the nuclear envelope, whereas the levels of Gcl localised to the nuclear membrane in the mutant *casp^lof^/Df* pole buds is significantly reduced. Furthermore, in the mutant embryos the bud extrusion is ineffective as assessed by the position of the Gcl^+ve^ buds (see arrowheads, B4). Magnified images of individual pole buds from both the genotypes are shown (C,D). The difference in the total amount of Gcl at the nuclear membrane is readily apparent (*casp^lof^/Df* embryos; see arrowhead, D3). The nucleoplasmic levels of Gcl seem elevated in *casp^lof^/Df* embryos (see asterisk, D3) as compared to control pole buds (asterisk, C3). Nuclear defects can be observed in the PGC of *casp^lof^/Df* embryo (arrowhead, D1). (E) Signal to noise ratio (SNR) comparison of 20 PGCs each for both the genotypes. Unpaired *t*-test, ***$P<0.001$.

in the case of nuclear defects, centrosomes from mutant embryos started to show aberrant behavior starting NC6/7. Some of the more common phenotypes included incomplete or delayed separation, presence of 'orphan' centrosomes devoid of DNA and speckled appearance of fragmented centrosomal material that was labeled by the antibodies against γ-tubulin or centrosomin. These phenotypes were not seen in similarly aged control embryos. Importantly, aberrations were also noticed in the pole bud nuclei of *casp^lof^* embryos. Moreover, the nuclear division defects, including uneven chromosome segregation and kinetochore aberrations, may not necessarily be connected to centrosomal abnormalities. Altogether,

the early nuclear, as well as centrosomal division cycles, are highly aberrant in *casp^lof^* embryos.

Together our data show that nuclei and associated centrosomes show several morphological and behavioral aberrations in presyncytial *casp^lof^* embryos at NC 6-8, even before cortical nuclear migration is initiated. Furthermore, these data strongly argue that in *casp^lof^* embryos, defective nuclei and centrosomes enter the pole plasm and perhaps induce PGC loss, which is further aggravated by the reduction in pole plasm due to diminished Osk. This possibility is especially illuminating as loss of *osk* does not induce nuclear migration defects. Thus, both the nuclear migration defects and

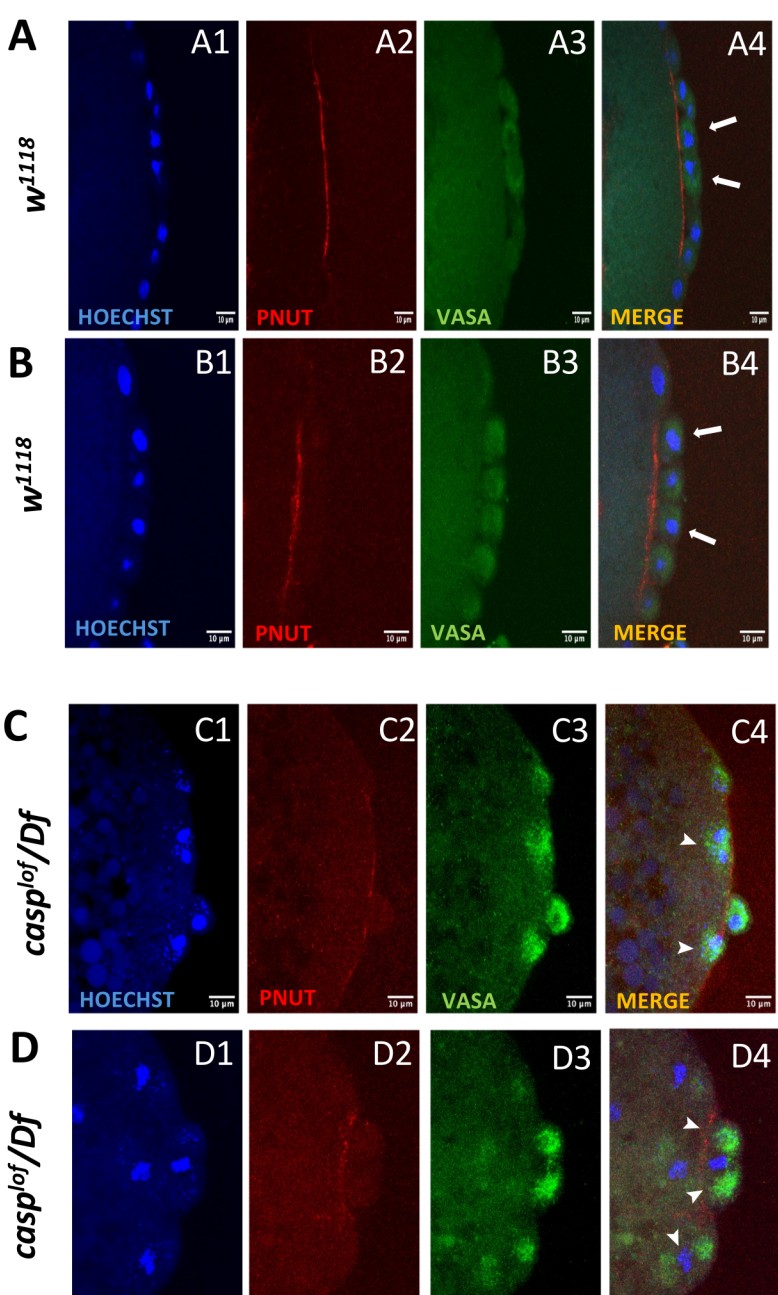

**Fig. 6. Casp regulates bud extrusion by influencing the actin cytoskeleton components responsible for contractile ring formation.** (A-D) Confocal maximum intensity projections of nuclear cycle (8/9) embryos derived from $w^{1118}$ (A,B) and $casp^{lof}/Df$ (C,D) females immunostained with Hoechst, Pnut, and Vasa antibodies. Pnut marks the actin cytoskeleton. $w^{1118}$ embryos (A4,B4) that show proper bud formation display accumulation of Pnut protein at the base before buds are about to pinch off (see arrows) to achieve precocious cellularization. However, in $casp^{lof}/Df$ embryos (C4,D4), the contractile ring formation seems incomplete and the band of Pnut staining at the base is discontinuous, which can be correlated with the incomplete bud protrusion from the somatic layer (see arrowheads).

aberrant centrosome migration seen in $casp^{lof}$ embryos are unlikely to be an outcome of reduction in Osk. Similarly, while reduction in Gcl, (an established target of Osk), likely contributes to PGC loss, it cannot explain these early phenotypes in $casp^{lof}$ embryos.

Altogether loss of pole buds (and PGCs) in $casp^{lof}$ embryos is likely induced by 1) defective nuclear/centrosomal migration and 2) reduction in Osk levels, which results in compromised germ plasm assembly. At present, the relative contributions of these two factors are not readily discernible, however.

### Both compartmentalization of somatic cytoplasm and pole bud formation are contemporaneous and employ an overlapping set of proteins

Bud formation and subsequent precocious cellularization is one of the distinguishing features of PGCs. This achieves germ plasm segregation leading to germline/soma distinction. It is adversely affected by loss of *casp*. Moreover, somatic cellularization is also disrupted in $casp^{lof}$ blastoderm embryos. Although cellularization starts at cycle 14, somatic compartmentalization is initiated in precellular blastoderm embryos. We wondered if the aberrant somatic cellularization and PGC formation defects seen in $casp^{lof}$ embryos are mechanistically connected. Thus, we turned our attention to the proteins needed for compartment formation in pseudocellular-cleavage stage embryos, which also contribute to budding of pole cells. We examined if Casp is also necessary during the formation of somatic compartments seen in early-stage pseudo-cleavage embryos.

We therefore sought to examine whether the organization of somatic cytoplasm commences at the same time when budding of pole cells takes place and, if so, does that organization depend on

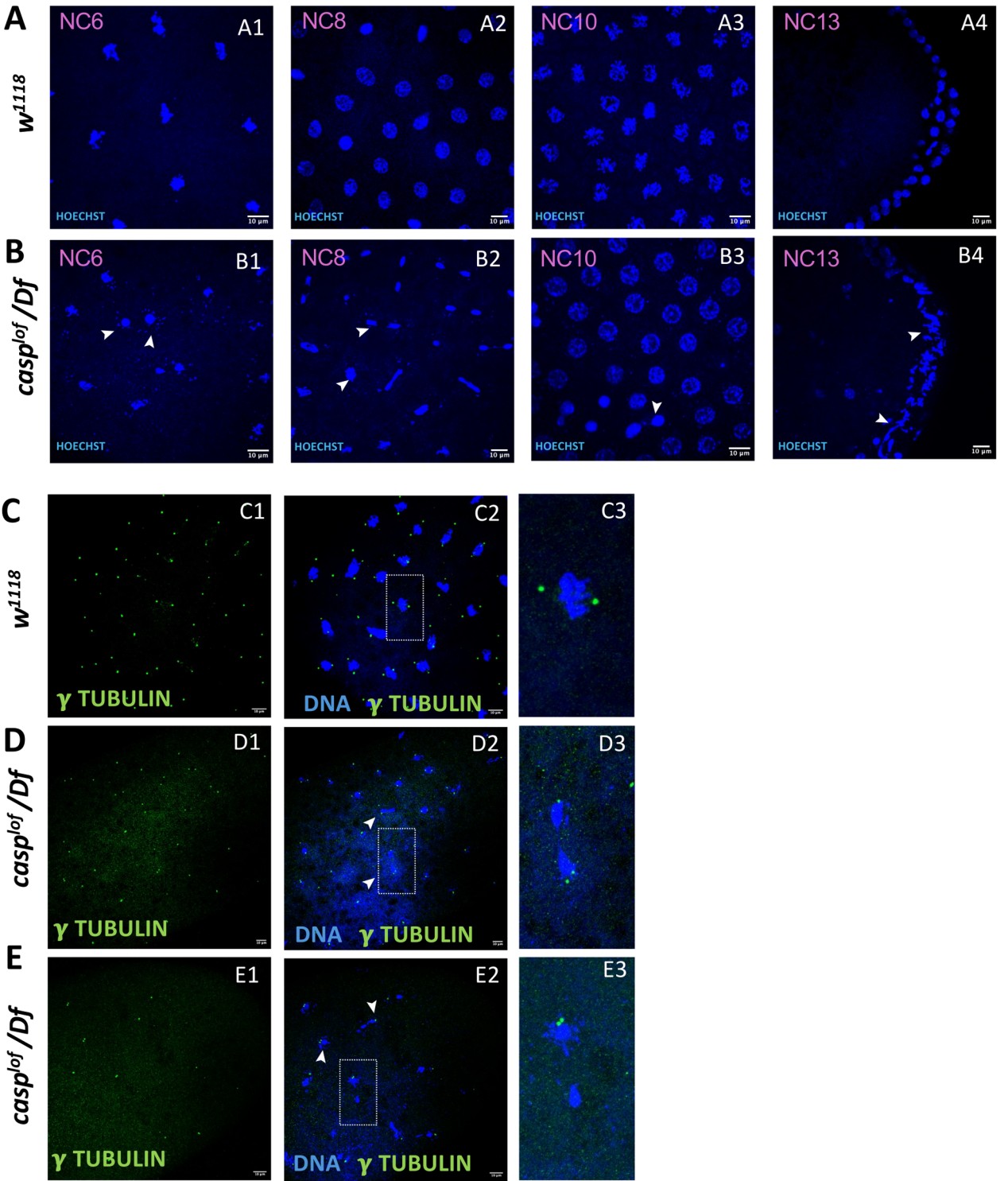

**Fig. 7. Casp regulates nuclear divisions and mitotic synchrony in pre-syncytial embryos.** (A,B) Nuclear cycle (6-13) embryos laid by $w^{1118}$ (A) and $casp^{lof}/Df$ (B) stained with Hoechst (marking the nuclei). In $casp^{lof}/Df$ embryos nuclear migration defects become apparent (B1-B4; see arrowheads) in the pre-syncytial nuclear cycles (B1-B3) even prior to cortical nuclear migration is typically initiated. Also note scattered dots labeled with Hoechst, which likely indicate fragmented or disintegrating DNA. (C-E) Immunostained pre-syncytial embryos of the same genotypes (C, $w^{1118}$) (D, E, $casp^{lof}/Df$) labeled with anti γ-tubulin antibodies to mark the centrosomes. Defective centrosomes are observed in $casp^{lof}/Df$ embryos (see arrowheads, D2, E2). C3, D3, E3 are the magnified images (3.5X) of the nuclei with centrosomes (C2, D2, E2) shown in the inset. The abnormalities include incomplete separation of duplicated centrosomes post-division or delayed migration.

Casp. In the control embryos, Pnut protein localization is structured in Cycle 9 onwards and the regular polygonal organization is apparent (Fig. 8A-C). To investigate if this behavior is unique to

Septin, similar antibody staining was performed using antibodies against the membrane-associated protein Discs large (Fig. 8D,E; Fig. S2).

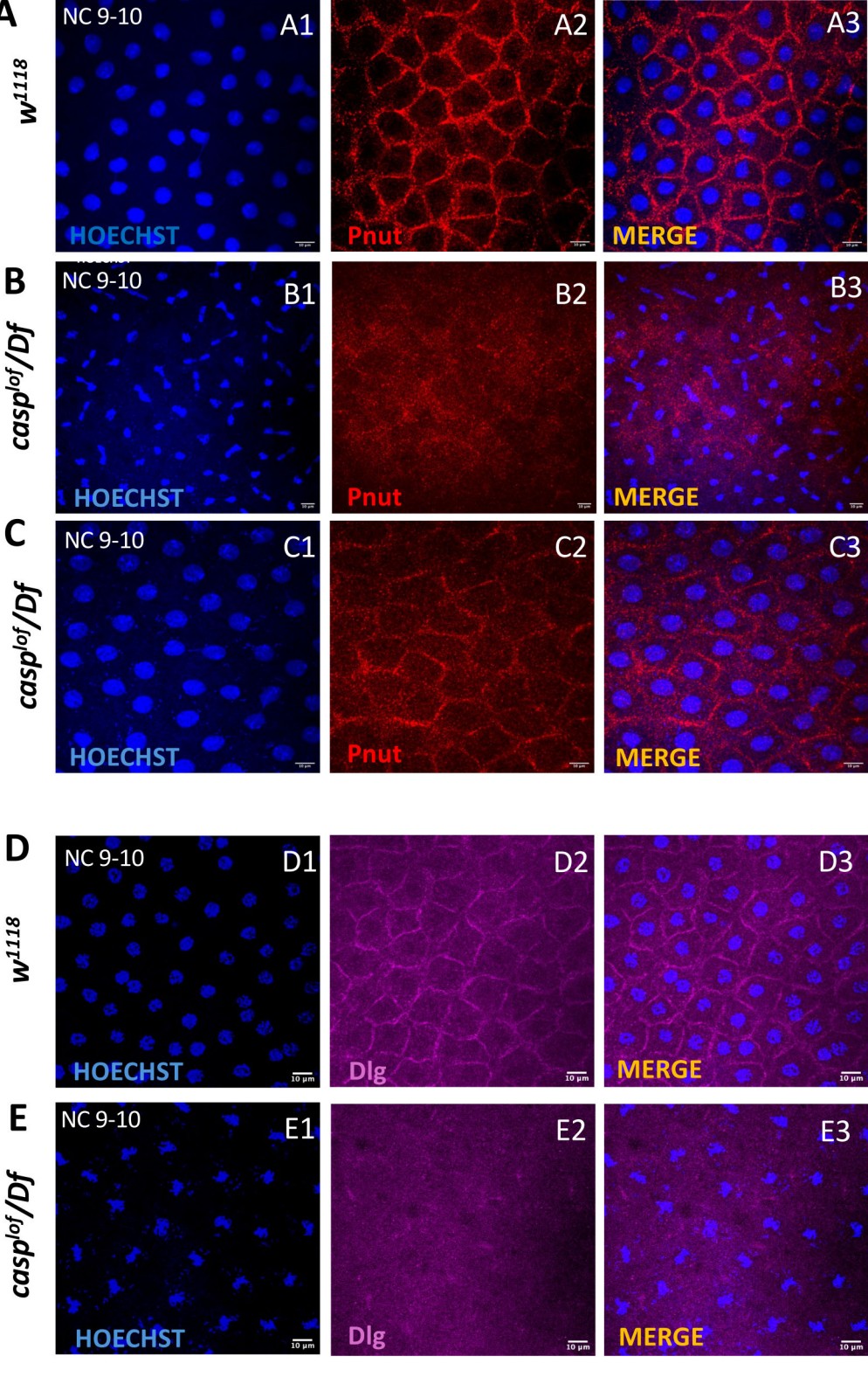

**Fig. 8. Initiation of somatic compartmentalization reflected in polygonality and its relationship with Casp activity.** (A-C) Nuclear cycle (9/10) $w^{1118}$ (A) and $casp^{lof}/Df$ (B,C) embryos immunostained with Pnut antibodies and Hoechst DNA dye. The somatic compartmentalization as indicated by the regular polygonal arrangement appears disrupted in $casp^{lof}/Df$ embryos (B2,C2) as compared to the control (A2). (D,E) Nuclear cycle (9/10) embryos of the same genotype immunostained with a membrane marker Dlg and labeled with Hoechst. During early nuclear cycles Dlg seems to outline the membranes of individual polygons that embryos are composed of. The regular spacing and structured pattern is lost in $casp^{lof}/Df$ embryos (E2) as compared to control embryos (D2).

Interestingly, similarly staged pre-syncytial blastoderm embryos between NC 9 and 10 show regularly spaced anti-Dlg specific staining. Furthermore, the pattern is disrupted in $casp^{lof}$ embryos (Fig. 8D,E; Fig. S2). These data argue that the somatic compartmentalization and pole bud formation are initiated around the same time and overlapping sets of protein contribute to the process underlying germline/soma distinction in young pre-syncytial embryos. Moreover, centrosome abnormalities correlate with uneven distribution of both the actin cytoskeletal components and membrane proteins in the somatic as well as germline compartment.

## DISCUSSION

Casp, in conjunction with its protein partner TER94, regulates PGC formation and their subsequent mitotic divisions in pre-syncytial

blastoderm embryos (Das et al., 2024). Furthermore, maternal overexpression of Casp alone results in excess number of PGCs. In this regard it resembles Gcl (Jongens et al., 1994, 1992). Like Gcl, Casp also influences even partitioning of germ plasm between early PGCs undergoing mitosis (Das et al., 2024; Lerit et al., 2017). In $gcl^{m-}$ mutant PGCs, this defect was attributed to the ability of Gcl protein to influence centrosome dynamics. Interestingly, $casp^{lof}$ embryos also display defective centrosome separation which correlates with inequitable distribution of germ plasm among daughter cells.

However, there are a few important distinctions between these two PGC determinants. Gcl is detected exclusively in the PGCs whereas Casp, while enriched in PGCs, is also uniformly distributed in the somatic cells. Accordingly, Gcl function is confined to regulating PGC numbers and fate, whereas Casp is necessary during nuclear divisions/migration and MBT. Consistently, $gcl^{m-}$ embryos develop into viable adults that are semi-fertile (Jongens et al., 1994). By contrast, $casp^{lof}$ embryos are partially viable and semi-fertile. Nonetheless, perhaps the most critical distinction relates to their regulatory relationship with Osk. $gcl$ is a target of Osk, and its transcript and protein levels are diminished in $osk$ mutant embryos. On the contrary, in $casp$ mutant embryos, Osk protein is reduced considerably. Our previous observations have argued that diminished levels of Osk can be attributed to downregulation of Smg levels/activity by Casp (Das et al., 2024).

Smg, influences $osk$ translation adversely (Siddiqui et al., 2024) and maternal loss of $smg$ leads to increase in PGC count that is comparable to $casp^{OE}$ embryos. Satisfyingly, loss of $casp$ led to increase in Smg. Thus, these data suggested a possible presence of a novel regulatory feedback loop between Osk, Casp and Smg activities in the young embryos. Consequently, it was proposed that while germ plasm assembly is a maternal process, it can be potentiated in the embryo to influence PGC numbers.

Our data show that Casp is needed to modulate both Osk and centrosome dynamics in the young embryo (Fig. 9). Supporting the idea, both Smg and Casp are barely detectable in the mature oocytes (Das et al., 2024; Siddiqui et al., 2024). Importantly, centrosome (and nuclear) migration in $osk$ embryos is comparable to wild type (Yohn et al., 2003). Thus, while pole plasm assembly and function

are controlled by Osk, centrosome dynamics, the other determinant of PGC formation, is Osk-independent. Of note, a germline clonal analysis based genetic screen did not yield any candidate that could regulate both Osk and nuclear migration (Yohn et al., 2003). By contrast, Casp is unique as it independently modulates both the determinants of PGC formation namely embryonic Osk levels and centrosome migration/ behavior (Fig. 9). This bi-modal functionality allows Casp to influence Osk activity and/or levels while the uniformly distributed Casp modulates centrosome dynamics in presyncytial as well as syncytial blastoderm embryos. Future experiments will focus on mechanistic underpinnings of how Casp accomplishes these two outcomes. It also remains to be determined how TER94, the protein partner of Casp (Tendulkar et al., 2022) in the germline (Thomson et al., 2008), contributes to this.

Lastly, our results suggest functionally analogous involvement of maternal regulators that are present both in the embryonic soma and PGCs. Casp is necessary in both compartments as it influences both pole cell budding and pseudo-cleavage embryo formation. While blastoderm embryo is cellularized only by NC 14, the somatic cytoplasm starts to display signs of physical and functional compartmentalization even earlier (Mavrakis et al., 2009). Subdivision of the cytoplasm is initiated as nuclei reach the cortex in syncytial blastoderm embryos. Interestingly, pole buds form even prior to NC 10. Localization patterns of Septin and Dlg indicate that somatic cytoplasm shows signs of polygonal compartments at the same time when bud formation is commenced. Moreover, both processes are similarly disrupted when $casp$ activity is compromised. Unlike control, in mutant embryos, somatic compartments do not appear to be enclosed completely. Similarly, pole buds do not pinch off efficiently either. It is intriguing that mutations in $dFMR$ and $ago2$ display similar phenotypes (Deshpande et al., 2005) (Deshpande et al., 2006). Like Casp, these proteins are present in both soma and PGCs. Germ plasm sequestration is a crucial step in establishing germline/soma distinction. As the ectopic expression of germline determinants is detrimental to somatic identity, the germline determinants are considered to be the more active participants in this process. That the determinants of zygotic genome activation (Zelda, CLAMP, etc.)

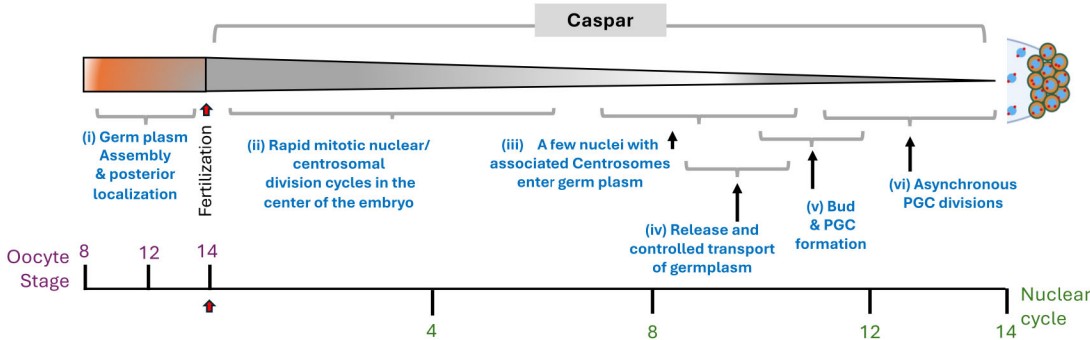

**Fig. 9. Casp influences PGC formation.** Casp participates in the process of PGC formation in a dual, i.e. Osk dependent and Osk-independent manner. a) It regulates pole plasm assembly by regulating Osk levels and thus it can modulate proper pole bud formation via cellular/cytoplasmic architecture, and mitotic divisions of the pole buds and the PGCs More importantly, it also controls nuclear and centrosomal division cycles even prior to pole bud formation during early NCs in an Osk independent manner. The schematic displays a timeline of PGC formation with critical events numbered i-vi. (i) Germ plasm is assembled pre-fertilization at the posterior cortex of the oocyte. (ii) Post-fertilization, the diploid nucleus undergoes 14 serial mitotic divisions without cytokinesis. The first six NCs take place in the center of the embryo giving rise to a 'nuclear cloud'. (iii) The nuclei, along with associated centrosomes start to migrate to the nuclear periphery by NC 10. A few (two to three) nuclei precociously migrate and enter posteriorly localized and anchored germ-plasm. The centrosomes induce the release of the anchored germ plasm from the posterior cortex. (iv,v) Microtubules emanating from the centrosomes engineer the germ plasm transport in collaboration with minus end motors. To achieve germline soma segregation, the germplasm is sequestered initially into pole buds (v), which eventually give rise to PGCs. (vi) PGCs undergo controlled mitosis, allowing even partitioning of the germ plasm among the daughter cells, which is facilitated by astral microtubules.

Biology Open

contribute to germline/soma distinction has, however, challenged this notion (Colonnetta et al., 2023). Contemporaneous sub-division of somatic cytoplasm as pole cell budding also argues in favor of an active role played by the somatic components during germline/soma segregation. While forced expression of germline determinants results in misspecification of embryonic somatic cells, the reverse possibility has not been examined carefully. Future experiments will test this idea by exploring possible non-autonomous influence of somatic factors on PGC determination and their mechanistic underpinnings. While PGCs from early embryos are not completely impervious to somatic signaling (Colonnetta et al., 2022, 2023; Deshpande et al., 2014; Pae et al., 2017), it may be worth exploring whether a calibrated communication between the soma and the PGCs in early embryos is critical for properly specifying both compartments. The biological significance of early partitioning of somatic cytoplasm in terms of patterning is well documented. Its functional implications for germline/soma distinction have remained unclear and await further examination.

## MATERIALS AND METHODS

### Fly husbandry and stocks
Flies were raised on standard cornmeal agar at 25°C unless stated otherwise. Bloomington *Drosophila* Stock Centre (BDSC) stock numbers for lines used in this study are BDSC_11373, *Casp^{lof}* (*w^{1118}; PBac{w^{+mC}=PB}casp^{c04227}*), BDSC_23691, *Casp Df* (*w^{1118}; Df(2R)BSC308/CyO*) and BDSC_4937 *nos Gal4* (*P{GAL4::VP16-nos.UTR}CG6325^{MVD1}*). The *UASp-Casp^{WT}* line is described in Das et al. (2024).

### Fixation, immunostaining, and imaging of embryos
Embryos were collected, dechorionated, fixed and stored in methanol/ethanol at −20°C using standard protocols as previously described (Das et al., 2024). For immunostaining, embryos were rehydrated by washing three times with 0.3% PBS-Triton X100 (PBST) for 15 min each. Embryos were then blocked in 2% bovine serum albumin (BSA) in 0.3% PBS-T for 1 h at room temperature. Embryos were incubated at 4°C overnight with primary antibodies diluted in 2% BSA in 0.3% PBST at the stated dilutions. Following four 15-min washes with 0.3% PBS-T, embryos were incubated in the appropriate secondary antibodies for 1 h at room temperature. The following antibodies were used: rat anti-α-Vasa, 1:1000; rabbit anti-Osk, 1:1000; mouse anti-γ-tubulin, 1:1000 (T6557, Sigma-Aldrich), mouse anti-Dlg, 1:25 (DSHB); rabbit anti-Gcl 1:500; mouse anti-Pnut (1:40). The following secondary antibodies were used, goat anti-mouse Alexa488/goat anti-rabbit Alexa568/goat anti-rat Alexa647/goat anti-rabbit Alexa647/goat anti-rabbit Alexa480, 1:1000 (DSHB). Embryos were washed three times with 0.3% PBS-T, and Hoechst (1:500 from a stock of 10 mg/ml) was added in the penultimate wash. Embryos were mounted using the Invitrogen SlowFade™ global antifade reagent and observed under a Zeiss multiphoton LSM 780 confocal and evident inverted microscope FV4000 with a 63X and 60X objective, respectively.

### RNA *in situ* hybridization
0-2 h-old embryos were collected in a sieve and dechorionated in 4% sodium hypochlorite for 90 s. After thorough washes with distilled water, embryos were fixed in a 1:1 heptane: 4% PFA solution for 20 min. The PFA layer was removed, and embryos in heptane were reconstituted with an equal volume of methanol. Embryos were devitellinized by vigorous shaking in the 1:1 heptane: methanol mixture. The heptane layer and the interphase containing non-devitellinized embryos were carefully removed. Devitellinized embryos in the bottom methanol phase were washed twice with methanol and stored at −20°C. Antisense digoxigenin-labeled RNA probes for *nos* was used and hybridization was carried out as previously described (Hegde et al., 2022; Tautz and Pfeifle, 1989). Anti-digoxigenin rhodamine antibody (Roche, 11207750910) was used at a concentration of 1:1000 to visualize *nos* mRNA. Images were acquired on a Zeiss multiphoton confocal LSM 780 confocal with a 63X objective.

### Quantification of microscopy images
Image analysis was performed using ImageJ. For fluorescence intensity measurement for *nos* mRNAs, all the z-stack slices were summed for each image. After careful background subtraction, an ROI encompassing the total *nos* mRNA spread was taken, and fluorescence intensity was measured. Each data point on the graph represents a biological replicate (fluorescence intensity of an individual embryo). In quantifying *nos* mRNA volume in the embryos, an ROI enclosing the posterior part of the embryo was drawn, which was kept consistent for all the images. Background subtraction in 3D throughout the stack was achieved using Gaussian blur 3D (X sigma=2, Y sigma=2 and Z sigma=2). Following proper thresholding, the volume of *nos* mRNA was examined using a 3D objects counter. To quantify the spread of Vasa, maximum intensity projections were used to compare the spread of Vasa in control and *casp^{lof}* embryos. Line intensity profile was measured and plotted using 75 µm line ROI starting from the posterior-most tip of the embryos. Each color on the graph represents one individual embryo. Gcl SNR quantification was performed using sum intensity projections for the stacks of individual PGCs. Three ROIs were drawn on the nuclear membrane to average the intensity of the Gcl signal of a particular PGC. The mean intensity of the Gcl signal in the PGC was then subtracted from the background, which was then divided by the standard deviation of the background to calculate SNR ratio. Ten PGCs were analyzed per embryo, and their SNR was plotted.

### Acknowledgements
Stocks obtained from the Bloomington *Drosophila* Stock Center (NIH P40OD018537) were used in this study. Flybase was an essential lookup and analysis resource, is funded by US NGHRI grant # U24HG013300 & US NSF grant # 2039324. We thank Paul Lasko and Trudi Schupbach for the Gcl and Vasa antibodies, respectively; Richa Rikhy and Amrita Roy for their helpful discussions; Snehal Patil and Yashwant Pawar for providing fly media and stock maintenance; the IISER Microscopy facility, Dr Santosh Podder, and Vijay Vittal, for their microscopy training and facility maintenance.

**Competing interests**
The authors declare no competing or financial interests.

**Author contributions**
Conceptualization: S.D., G.D., G.S.R.; Data curation: S.D.; Formal analysis: S.D., A.E.R., K., G.D., G.S.R.; Funding acquisition: G.S.R.; Investigation: S.D., A.E.R., K., G.S.R.; Methodology: S.D., A.E.R., K., G.D., G.S.R.; Project administration: G.S.R.; Supervision: G.D., G.S.R.; Validation: A.E.R., K.; Visualization: G.D., G.S.R.; Writing – original draft: G.D., G.S.R.; Writing – review & editing: S.D., A.E.R., K., G.D., G.S.R.

**Funding**
Pratiksha Trust Extra-Mural Support for Transformational Ageing Brain Research grant EMSTAR/2023/SL03 to G.R., facilitated by the Centre for Brain Research (CBR), Indian Institute of Science, Bangalore to G.R.; Indian Institute of Science Education and Research (IISER), Pune for intramural support to G.R.'s research group. G.D.'s visits to IISER, Pune (2023, 2024) were supported by the Ministry of Education (MoE), Scheme for the Promotion of Academic & Research Collaboration (SPARC), grant ID SPARC-1587, managed by IIT Kharagpur. The IISER *Drosophila* media and stock centre are supported by the National Facility for Gene Function in Health and Disease (NFGFHD) at IISER Pune, which in turn is supported by an infrastructure grant from the Department of Biotechnology, Ministry of Science and Technology (DBT), India (BT/INF/22/SP17358/2016). S.D. is supported by an EMSTAR Project Associateship; A.E.R. is supported by a KVPY/INSPIRE government of India fellowship for undergraduate students. K. is a graduate student supported by a research fellowship from the Council of Scientific and Industrial Research (CSIR), India. Open Access funding provided by Indian Institute of Science Education & Research Pune. Deposited in PMC for immediate release.

**Data and resource availability**
All relevant data and details of resources can be found within the article and its supplementary information.

**First Person**
This article has an associated First Person interview with the first author of the paper.

**Peer review history**
The peer review history is available online at https://journals.biologists.com/bio/lookup/doi/10.1242/bio.062119.reviewer-comments.pdf

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
