## [Peer Review File · Biology Open]

Caspar modulates both Oskar-dependent and Oskar-independent activities during embryonic germ cell formation and fate determination

Subhradip Das, Adheena Elsa Roy, Kanika K, Girish Deshpande and Girish S. Ratnaparkhi
DOI: 10.1242/bio.062119

Editor: Tristan Rodríguez

Review timeline

Original submission:	5 May 2025
Editorial decision:	28 May 2025
First revision received:	16 June 2025
Accepted:	18 June 2025

Original submission

First decision letter

MS Title: Caspar modulates primordial germ cell fate both in Oskar-dependent and Oskar-independent manner

Authors: Subhradip Das; Adheena Elsa Roy; Kanika K; Girish Deshpande; Girish S Ratnaparkhi
Article Type: Research Article

Dear Dr Ratnaparkhi,

I have now received all the referees' reports on the above manuscript, and have reached a decision. I am sorry to say that the outcome is not a positive one. The referees' comments are appended below, or you can access them online: please go to.

As you will see, while the reviewers' agree that the experiments presented are well done, they also raise significant concerns about the study. In particular, there is lack of evidence on actually testing the model proposed in the study on the role of Caspar and SCF component in mediating the proposed function, and a clear lack of characterization of the mitotic function of the mutant. Having looked at the manuscript myself, I agree with their views, and I must therefore, reject your paper.

Reviewer 1

This manuscript is a follow up to <https://elifesciences.org/articles/98584#s3> where the authors identified Caspar, an orthologue of Fas associated factor 1, as required for multiple aspect of early embryo development, including aberrant centrosome behavior, gastrulation and pole cell formation. That paper speculated that the effect on pole cell formation may be due to Caspar possible involvement in tuning the levels of Smaug, a translational repressor of Oskar and other pole plasm components. This model was based on observing high levels of Smaug and low levels of Oskar in Caspar mutants and on the fact that Caspar orthologues interact with SCF (Skp Cullin and F-box (SCF) containing complex) which has been implicated in marking Smaug for degradation.

The new paper does not directly test this hypothesis. Instead the authors examine in great detail how Caspar mutations affect the level and distribution of a number of germ plasm proteins. While these observations are a great foundation, they are not sufficient on their own to build a mechanistic understanding of Caspar function. The authors conclude that Caspar affect centrosome behavior and germ plasm assembly initially independently and that the centrosome defects further exacerbates the germ plasm defects by interfering with proper distribution of germ plasm to pole cells. This is a reasonable model but does not add much to what was already inferred in the prior study. It is unfortunate that the authors do not follow up on the possibility that Caspar's various function are linked to SCF-dependent degradation of multiple proteins in early embryos, as suggested in their prior study. In Summary, this paper provides additional information as to the germ plasm defects of Caspar mutants, but fails to provide definitive insights into Caspar function.

Reviewer 2

SUMMARY OF THE ADVANCE MADE IN THIS PAPER AND ITS POTENTIAL SIGNIFICANCE TO THE FIELD

This manuscript by Das et al. reports the detailed characterization of Caspar in PGC formation in *Drosophila*. Following up on their earlier work, which identified Caspar as being required for PGC formation, this work now demonstrates that Caspar functions in two pathways--- Oskar-independent and -dependent manner. For the *osk*-independent branch, they show that Caspar functions to regulate centrosome dynamics.

Specifically, they show 1) PGC number is decreased in *casp* mutant, 2) *nos* mRNA is reduced in *casp* mutant. 3) although *osk* and *nos* are reduced, *Vasa* spreads across the posterior half. 4) This abnormal anchoring of multiple components (*nos* RNA, *Vasa* protein) is due to abnormal centrosome function. 5) *Gcl* protein is reduced in *casp* mutant. 6) contractile ring formation is defective in *casp* mutant PGCs (likely due to *Gcl* loss).

Overall, the experiments are thoroughly done, and lots of phenotypes are described, but it leaves the impression that the paper did not get to the bottom of the phenomenon. In particular, *casp* mutant seems to have a clear mitotic defect, which appears to be missed by the authors (only described as 'nuclear fragmentation', which is a misdiagnosis, I believe). And this phenotype appears to be common between germ cells and somatic cells. Without a better characterization of the phenotype, it will be difficult to conclude that *casp*'s function in PGC formation is unique to germ cell biology or it is a downstream outcome of a more general cell division defect.

SUGGESTIONS TO AUTHORS

-The abstract is unclear as to what was already known and what are new discoveries.

-They say that *Casp*'s function is almost exclusively maternal. Is this true with all phenotypes they describe? (not just the initial PGC number, but also later migration after MZT). Especially because they describe multiple functions of *Casp*, it would be important to show which function is maternal vs. zygotic. In particular, the results with mutant mothers crossed to wild-type fathers would be very informative.

-line 256 'trace amount of *nos* mRNA is spread in the rest of the embryo'. This description is incorrect. I believe it is shown that the majority of *nos* mRNA is spread across the embryo, but it is only 'more concentrated' in the posterior pole, and is exclusively translated at the posterior pole. Please correct.

-*Vasa*'s spread localization in the *casp* mutant requires a better quantification. Can they use *Vasa*-GFP to confirm the results with the *Vasa* antibody? This confirmation is important, particularly because it is the key result to justify the later experiments described in the paper. Also I noted that in Fig5B, *Vasa* seems to be simply much less in *casp* mutant, instead of being spread.

-in Fig4, they describe abnormal distribution of *Vasa* protein along the dividing germ cells, inferring that this defect is due to abnormal centrosome function. However, in addition to these defects, mitosis is clearly compromised (clear sister chromatid segregation defects). Such a defect is not

necessarily attributable to centrosome function. It is also possible that the kinetochore function is abnormal. Please describe this defect more thoroughly. (Fig7 describes some of this, but it is not 'nuclear fragmentation' as is described. This is a clear mitotic cell division defect resulting from an inability to correctly separate sister chromatids.

First revision

Author response to reviewers' comments

Reviewer 1: (1) This manuscript is a follow up to <https://elifesciences.org/articles/98584#s3> where the authors identified Caspar, an orthologue of Fas associated factor 1, as required for multiple aspect of early embryo development, including aberrant centrosome behavior, gastrulation and pole cell formation. That paper speculated that the effect on pole cell formation may be due to Caspar possible involvement in tuning the levels of Smaug, a translational repressor of Oskar and other pole plasm components. This model was based on observing high levels of Smaug and low levels of Oskar in Caspar mutants and on the fact that Caspar orthologues interact with SCF (Skp Cullin and F-box (SCF) containing complex) which has been implicated in marking Smaug for degradation.

(2) The new paper does not directly test this hypothesis. Instead the authors examine in great detail how Caspar mutations affect the level and distribution of a number of germ plasm proteins. While these observations are a great foundation, they are not sufficient on their own to build a mechanistic understanding of Caspar function. The authors conclude that Caspar affect centrosome behavior and germ plasm assembly initially independently and that the centrosome defects further exacerbates the germ plasm defects by interfering with proper distribution of germ plasm to pole cells. This is a reasonable model but does not add much to what was already inferred in the prior study. It is unfortunate that the authors do not follow up on the possibility that Caspar's various function are linked to SCF-dependent degradation of multiple proteins in early embryos, as suggested in their prior study. In Summary, this paper provides additional information as to the germ plasm defects of Caspar mutants, but fails to provide definitive insights into Caspar function.

Thank you for the feedback. We would like to point out that the main theme of this manuscript was (and still is) to highlight Caspar's dual-influence on primordial germ cell (PGC) formation, post-fertilization. Our earlier data (Das et al, 2024) indicated that Caspar influences two functionally separable regulators of PGC formation, (1) Levels and/or activity of Oskar protein which is necessary and sufficient to assemble germplasm in the oocyte and, (2) The dynamics of centrosome migration and their subsequent entry into the posteriorly anchored germplasm which induces the controlled release and transmission of germ plasm components. Both the processes are critical for PGC formation and germline soma distinction. Our data also emphasizes that Caspar, in the PGCs, regulates Oskar. As a consequence Caspar can directly or indirectly controls targets (Vasa protein, *nos* RNA etc.) and regulators (Smaug) of Oskar. We proposed that this is achieved possibly by regulating degradation of one or multiple germ plasm proteins. As reviewer #1 points out, documenting the effects of Caspar (*lof/gof*) is an excellent foundation for future studies, and, in our opinion, are essential for further detailed mechanistic exploration. Thus the current data albeit being descriptive are valuable.

Nonetheless, we agree with the reviewer's assessment that further mechanistic work, along the lines of Caspar's role in SCF-mediated degradation (Das et al., 2024, *eLife*), will be the next crucial step. We are following up on these experiments and expect to uncover molecular aspects of Caspar-mediated PGC regulation. We believe that our extensive phenotypic analysis of Caspar function in the embryonic PGCs is relevant and will be of sufficient interest to the diverse audience of 'Biology Open'. Moreover, consistent with its presence in the soma and the PGCs, we also show its role in early embryonic somatic compartment. This will be useful in tracing its subsequent role in MBT which we discussed in our earlier publication (Das et al, 2024).

Reviewer 2: (1) This manuscript by Das et al. reports the detailed characterization of Caspar in PGC formation in *Drosophila*. Following up on their earlier work, which identified Caspar as being required for PGC formation, this work now demonstrates that Caspar functions in two pathways--- Oskar-independent and Osk-dependent manner. For the ask-independent branch, they show that Caspar functions to regulate centrosome dynamics.

We thank the reviewer for supporting our basic premise that Caspar has both Oskar-dependent and Oskar-independent roles during PGC formation and specification. The argument is based on our research, published last year (Das et al., 2024, *Elife*), and the current study. The Oskar-independent role relates to Caspar's effect on centrosome dynamics, documented extensively in the current study, while the Oskar-dependant role relates to Caspar's influence on germ-plasm components, including Oskar itself.

(2) Specifically, they show 1) PGC number is decreased in casp mutant, 2) nos mRNA is reduced in casp mutant. 3) although osk and nos are reduced, Vasa spreads across the posterior half. 4) This abnormal anchoring of multiple components (nos RNA, Vasa protein) is due to abnormal centrosome function. 5) Gcl protein is reduced in casp mutant. 6) contractile ring formation is defective in casp mutant PGCs (likely due to Gcl loss).

Reviewer #2 describes the different datasets collected and presented in the manuscript, which are used to underscore the Oskar-Independent and Oskar-Dependent aspects of Caspar function. Many of the phenotypes are striking, with quantification used to highlight statistical significance.

(3) Overall, the experiments are thoroughly done, and lots of phenotypes are described, but it leaves the impression that the paper did not get to the bottom of the phenomenon. In particular, casp mutant seems to have a clear mitotic defect, which appears to be missed by the authors (only described as 'nuclear fragmentation', which is a misdiagnosis, I believe). And this phenotype appears to be common between germ cells and somatic cells. Without a better characterization of the phenotype, it will be difficult to conclude that casp's function in PGC formation is unique to germ cell biology or it is a downstream outcome of a more general cell division defect.

We agree with Reviewer #2, that we need to elucidate molecular underpinnings of Caspar function. This is in progress. In particular, we are analysing the functional relationship between Smaug and Caspar to understand their opposite influence (Das et al., 2024 *eLife*; Siddiqui et al., 2024, *Science Advances*) on Oskar activity/levels and germline assembly and transmission via centrosome function. We have started exploring Caspar-mediated degradation of target proteins essential for PGC formation and specification. However, we believe that analysis presented in our current manuscript will prove sufficiently valuable to the general audience interested in mechanisms underlying germline-soma distinction.

SUGGESTIONS TO AUTHORS

-The abstract is unclear as to what was already known and what are new discoveries.

The abstract highlights the discoveries (points #1 - #6 as stated by reviewer in comment (2) above). Points #4-#6 are examined in detail in this study.

-They say that Casp's function is almost exclusively maternal. Is this true with all phenotypes they describe? (not just the initial PGC number, but also later migration after MZT). Especially because they describe multiple functions of Casp, it would be important to show which function is maternal vs. zygotic. In particular, the results with mutant mothers crossed to wild-type fathers would be very informative.

This is a very thoughtful suggestion. As far as its role during PGC division, Caspar function appears to be exclusively maternal. This was documented in our previous study (Das et al.,

2024, eLife). However we will assess if the maternal (Regulation of Oskar levels and germ plasm assembly) v/s embryonic functions (PGC division and centrosome dynamics) of Caspar can be uncoupled.

-line 256 'trace amount of nos mRNA is spread in the rest of the embryo'. This description is incorrect. I believe it is shown that the majority of nos mRNA is spread across the embryo, but it is only 'more concentrated' in the posterior pole, and is exclusively translated at the posterior pole. Please correct.

The sentence has been suitably modified to indicate that even in wild type embryos, less than ~5% of total nos RNA is tightly anchored to the posterior cortex and the rest is spread in the posterior half.

-Vasa's spread localization in the casp mutant requires a better quantification. Can they use Vasa-GFP to confirm the results with the Vasa antibody? This confirmation is important, particularly because it is the key result to justify the later experiments described in the paper. Also I noted that in Fig5B, Vasa seems to be simply much less in casp mutant, instead of being spread.

The Vasa antibody is an excellent reagent, at par with the best antibodies generated in flies. We do not expect Vasa-GFP to significantly alter our conclusions or interpretation of the data, especially in light of autofluorescence observed in early embryos which may cloud the quantification of the GFP signal. The Vasa levels appear less because of the fewer PGC's in *Casp^{lof}/Df* embryos (Fig 5B).

-in Fig4, they describe abnormal distribution of Vasa protein along the dividing germ cells, inferring that this defect is due to abnormal centrosome function. However, in addition to these defects, mitosis is clearly compromised (clear sister chromatid segregation defects). Such a defect is not necessarily attributable to centrosome function. It is also possible that the kinetochore function is abnormal. Please describe this defect more thoroughly. (Fig7 describes some of this, but it is not 'nuclear fragmentation' as is described. This is a clear mitotic cell division defect resulting from an inability to correctly separate sister chromatids.

The reviewer's observation is on point. We have edited the text to better describe the phenotype (See pg. 22, line 639-642). It should be pointed out that chromosome segregation and centromere defects will need careful attention and thorough analysis.

Second decision letter

MS ID#: bio.062119R1

MS TITLE: Caspar modulates both Oskar-dependent and Oskar-independent activities during embryonic germ cell formation and fate determination

AUTHORS: Subhradip Das; Adheena Elsa Roy; Kanika K; Girish Deshpande; Girish S. Ratnaparkhi

I am happy to tell you that your manuscript has been accepted for publication in Biology Open, pending our standard publication integrity checks. It was accepted on 18 June 2025.